# Effect of pH, Norepinephrine and Glucose on Metabolic and Biofilm Activity of Uropathogenic Microorganisms

**DOI:** 10.3390/microorganisms11040862

**Published:** 2023-03-28

**Authors:** Nadezhda Ignatova, Alina Abidullina, Olga Streltsova, Vadim Elagin, Vladislav Kamensky

**Affiliations:** 1Department of Epidemiology, Microbiology and Evidence-Based Medicine, Privolzhsky Research Medical University, 603104 Nizhny Novgorod, Russia; 2Department of Urology, Privolzhsky Research Medical University, 603104 Nizhny Novgorod, Russia; 3Institute of Experimental Oncology and Biomedical Technologies, Privolzhsky Research Medical University, 603104 Nizhny Novgorod, Russia

**Keywords:** biofilms, pH, norepinephrine, glucose, biomass, matrix, metabolism, *E. coli*, *S. aureus*, *Kl. pneumoniae*, *Ps. aeruginosa*, *E. faecalis*

## Abstract

Urinary tract infection (UTIs) aremainly caused by a number of anatomical and physiological dysfunctions, but there are also some iatrogenic factors, including the use of certain medications, that contribute to the development of UTIs. The virulence of bacteria that colonize the urinary tract may be modified by pH and by the presence of soluble substances in urine, such as norepinephrine (NE) and glucose. In this work, we studied the influence of NE and glucose across a range of pHs (5, 7, 8) on the biomass, matrix production and metabolism of uropathogenic strains of *Escherichia coli*, *Pseudomonas aeruginosa*, *Klebsiella pneumoniae*, *Staphylococcus aureus* and *Enterococcus faecalis*. We used Congo red and gentian violet to stain the extracellular matrix and biomass, respectively, of biofilms. The optical density of staining of the biofilms was measured using a multichannel spectrophotometer. The metabolic activity was analyzed by MTT assay. It was shown that NE and glucose stimulate biomass production both in the Gram-negative and Gram-positive uropathogens. The metabolic activity in the presence of glucose was higher at pH 5 for *E. coli* (in 4.0 ± 0.1 times), *Ps. aeruginosa* (in 8.2 ± 0.2 times) and *Kl. pneumoniae* (in 4.1 ± 0.2 times). Matrix production of *Kl. pneumoniae* increased under NE (in 8.2 ± 0.2 times) and in the presence of glucose (in 1.5 ± 0.3 times). Thus, NE and glucose in urine may lead to persistent UTI under patient stress and in the case of metabolic glucose disorders.

## 1. Introduction

Urine is a biological fluid which reflects the functional integrity of the body, so changes in its biochemical composition are some of the first indicators of possible pathological and physiological changes. One of the most important indicators in urine analysis comes from the determination of its pH. Since urine is practically devoid of mechanisms for maintaining homeostasis, the hydrogen index responds to any changes in the body, making pH a good biomarker [1]. Urine pH is slightly acidic (between 5.0 and 6.0), although normal values can range from pH 4.5 to 8.0 [2]. This wide range of pH values can be explained by the possibility of physiological factors. For example, the value of urine pH is reduced by a diet rich in animal protein [3,4,5], and by excessive consumption of sodium salts with food. Conversely, a diet rich in vegetables and fruits, the consumption of an adequate amount of water, and of calcium-rich foods contribute to an increase in pH [3,5]. pH also changes in pathological conditions. An increase in the hydrogen index above 7 is observed in metabolic and respiratory alkalosis [6], in chronic renal failure [7], hyperfunction of the parathyroid gland [8], with the administration of certain medications [9], in the case of neoplasms of the urinary system, and in urinary tract infections associated with microorganisms that break down urea [10]. A decrease in pH below 4 is associated with metabolic and respiratory acidosis [6], diabetes mellitus, obesity [11], and with taking some medications [12].

Interestingly, the pH value affects the process of stone formation. For example, when the pH is below 5.5, the solubility of uric acid increases, leading to the formation of uric acid crystals that can act as heterogeneous cores for calcium oxalate crystals. A pH above 6.0 sharply reduces the solubility of calcium phosphate, which, in turn, leads to the formation of calcium phosphate crystals that can act as heterogeneous nuclei for calcium oxalate crystals [13]. From this it follows that pH shifts in either direction can lead to the formation of uro- and nephrolithiases, which are often further complicated by the presence of infections [14].

Norepinephrine is one of the components of urine, normally present in small concentrations. The reference values range from 20 to 40 micrograms per day and depend on conditions such as stress, physical activity or the development of pheochromacytoma [15]. That norepinephrine produced by macroorganisms can affect microorganisms is well known [16], however, the issue of the changing activity of catecholamine action on bacteria at different pH values remains poorly understood.

In healthy people, urine is poor in the nutrients such as proteins and carbohydrates, that necessary for bacterial growth, but it is contain various soluble salts. In addition, native urine is a multicomponent system, which may contain some hormones such as estradiol [17], adrenalin [18] and drug metabolites [19]. The urine of each patient is an individual mixture of salts and metabolic products. It should be noted that each component of native urine may affect on experimental results. In order to avoid this phosphate-buffered saline (PBS) as an analogue of urine with adding one type of components was used. There is normally an absence of glucose in urine; if present, it is another important biomarker of pathology. Glucosuria is an increase in glucose in fresh urine of more than 0.25 mg/mL, observed as a result of exceeding the renal threshold (0.9–2.0 mmol/min) in the blood (for example, in diabetes mellitus) or due to impaired reabsorption in the kidneys [20]. There may also be a physiological increase in glucose in the urine with excessive carbohydrate content in the diet [21].

Changes in the presented parameters of urine, as mentioned above, can be both physiological and pathological, which significantly increases the frequency of occurrence of these deviations in laboratory tests. Since changes in urine constants can potentially affect uropathogens, it is relevant to study the effect of pH and biochemical changes on microorganisms.

## 2. Materials and Methods

### 2.1. Bacterial Strains and Cultivation

The studies were carried out on 12 strains of *Escherichia coli*, 13 strains of *Staphylococcus aureus*, 7 strains of *Pseudomonas aeruginosa*, 7 strains of *Klebsiella pneumoniae* and 6 strains of *Enterococcus faecalis* isolated from urine samples of patients with uroinfections undergoing treatment at the Nizhny Novgorod Regional Clinical Hospital named after N.A. Semashko. The species identity was determined by matrix assisted laser desorption ionization-time of flight mass spectrometry (MALDI ToF Autoflex speed, Bruker Daltonik GmbH, 28359 Bremen, Germany). Routine cultivation was carried out using nutrient agar (24 h, 37 °C). Nutrient broth was used as a liquid medium. Inoculated broth was cultivated for 16 h at 37 °C with continuous rotation at 250 rpm. For all experiments, overnight cultures of bacteria on a log phase of growth were diluted to 0.5 McFarland standard (1 × 10^8^ CFU/mL).

### 2.2. pH and Soluble Components of Urine

In experiments we used a phosphate-buffered saline (PBS) as a liquid medium, which imitates basic conditions similar to those in urine. PBS is an isotonic solution commonly used in biological research. It is a pH-adjusted (7.4) blend of disodium hydrogen phosphate, sodium chloride, potassium chloride and potassium dihydrogen phosphate. Due to highly complicated composition and strong variation between persons native urine were not used for experiments. Thus, it might be difficult to interpret results of experiments when native urine used as a medium.

All experiments were carried out at three pH values (5.0; 7.0; 8.0) corresponding to the values within the normal range found in urine [2]. Indicator test strips for semi-quantitative determination of urine pH Uri-pH (Biosensor AN LLC, Moscow Region, Russia) were used for determination of the pH value.

Two hundred microliters of 0.1 M HCl were added to 20 mL of PBS to adjust pH at 5.0. The pH 8.0 was set by adding 50 µL of 5 M NaOH to 27 mL of PBS. The studied pH level was set at the beginning of an experiment and was measured after cultured period of 24 h at 37 °C.

Glucose was added to PBS at a final concentration of 0.5 mg/mL, which is the upper limit of the normal reference values in urine [20] and norepinephrine (NE, Laboratoire Aguettant, Lyon, France) solution in PBS to a concentration of 0.052%, which corresponds to the mean physiological level of the catecholamine in the urine of healthy people as described previously [15]. The bacterial suspension with glucose solution or NE solution was shaken thoroughly before use in assays.

### 2.3. Effects on the Growth of Bacterial Biomass

Bacteria forming biofilms were cultured in polystyrene 96-well plates in PBS (pH 5.0; 7.0; 8.0) starting at a bacterial concentration of 1 × 10^8^ CFU/mL. To analyze the effect of NE on the growth of the bacterial biomass, it was added (0.052%) to experimental wells, while the control ones were left hormone free [16]. To analyze the effect of glucose on the growth of bacterial biomass, it was added (0.5 mg/mL) to different wells, while the control ones were left glucose free. The plates containing the suspensions were cultured for 24 h at 37 °C. After incubation, the biofilms were washed three times using phosphate buffered saline (PBS), fixed with 96% ethyl alcohol for 15 min, and then stained with 0.1% gentian violet solution (3 min). Next, the dye was eluted using 96% ethyl alcohol with constant shaking (10 min) before the optical density was measured using a multichannel spectrophotometer at a wavelength of 570/630 nm.

### 2.4. Matrix Production Assay

To analyze the effect of NE on matrix production, the biofilms were grown for 24 h in the presence of the hormone or glucose in polystyrene 96-well plates. The generated biofilms were washed three times with PBS and stained with Congo red for 15 min. The staining solution, prepared in PBS, contained 1% of Congo red and 10% Twin 80. Next, the dye was eluted using 96% ethyl alcohol with constant shaking (10 min) before the optical density was measured using a multichannel spectrophotometer at a wavelength of 490/540 nm [22].

### 2.5. Metabolic Activity Assay

A 3-(4,5-dimethylthiazol-2-yl)-2,5-diphenyltetrazolium bromide (MTT) reduction assay was used for detection of alterations in metabolic activity. Viable bacterial cells convert MTT into a purple colored formazan product with an absorbance maximum at 570 nm. The biofilm-forming bacteria were cultured in polystyrene 96-well plates on nutrient broth at a bacterial concentration of 1 × 10^8^ CFU/mL. After 24 h the biofilms were rinsed three times using PBS to remove planktonic bacteria. Then 100 µL of MTT solution (0.5 mg/mL) was added to each well. After 3 h incubation at 37 °C, the solution was replaced with 100 µL of dimethyl sulfoxide (DMSO) to dissolve the formazan crystals. The optical density of the resulting solutions was measured using a microplate reader at a wavelength of 570 nm and with a reference wavelength of 630 nm [16].

### 2.6. Statistical Analysis

The values of the optical densities are presented as means (M) and standard deviations of the mean (±SD) and calculated for all strains of each species. Statistical analysis was performed with Statistica 10 (StatSoft. Inc., Tusla, OK, USA). The nonparametric Mann-Whitney U-test was used. *p*-values ≤ 0.05 were considered to be statistically significant.

## 3. Results

### 3.1. Biomass Production of Uropathogenic Microorganisms

The presence of norepinephrine stimulated an increase in the biomass of all the studied uropathogens. However, depending on the initial pH of the liquid medium, we observed different degrees of biomass formation for individual bacterial species. Thus, more active stimulation of biomass formation in the presence of norepinephrine was noted at pH 7 and pH 8 for *E. coli* (in 4.1 ± 0.2 times), *Kl. pneumoniae* (in 3.2 ± 0.1 times and in 1.9 ± 0.2 times), *Ps. aeruginosa* (in 3.1 ± 0.3 times and in 2.8 ± 0.2 times) and *S. aureus* (in 3.0 ± 0.3 times and in 3.4 ± 0.2 times) (Figure 1). For *E. faecalis*, the growth was greater the more acidic the medium (pH 5–7) compared with *S. aureus* (Figure 1). We measured final pH (in 24 h incubation) for all species and noticed a shift to alkaline range (pH 8) after incubation with NE at initial pH 7. Other samples as well as control ones maintained the initial pH.

The presence of glucose in the medium did not always stimulate the growth of biomass of the uropathogens. Basically, the presence of glucose was important the more acidic the medium pH 5. This dependence was noted for all the studied uropathogen species, but especially for *Kl. pneumoniae* (in 8.3 ± 0.3 times), *Ps. aeruginosa* (in 2.2 ± 0.2 times), *E. coli* (in 1.3 ± 0.1 times) and *S. aureus* (in 1.4 ± 0.2 times) (Figure 2). Thus, if the urine has an acidic pH of 5, then the presence of glucose stimulates the growth of these uropathogens. In the alkaline medium (pH 8), significant stimulating effect of glucose on biomass formation was noted only for *E. coli* (in 1.4 ± 0.2 times) and *E. faecalis* (in 1.3 ± 0.3 times) (Figure 2).

We measured final pH (in 24 h incubation) for all species in glucose presence and noticed the shift of pH level to acid range (pH 5 to pH 4; pH 8 to pH 7). Control samples maintained the initial pH. Thus, we showed, that presence bacteria in PBS didn’t change pH, but adding NE and glucose may lead to shift of pH.

### 3.2. Matrix Production of Uropathogenic Microorganisms

Activation of matrix production under the action of norepinephrine was noted for the *Kl. pneumoniae* strains. The maximum stimulating effect was noted at pH 7 (in 8.2 ± 0.2 times) (Figure 3). No significant stimulation of matrix production was found for the other species. It is worth noting that for Gram-positive pathogens such as *S. aureus* and *E. faecalis*, the presence NE reduced matrix production (Figure 3).

The presence of glucose in the medium stimulated the production of matrix biomass only in *Kl. pneumoniae* strains regardless of the pH of the medium, however, Figure 4 also indicates significance of stimulation at pH 5 (in 1.5 ± 0.3 times) and at 8 pH (in 3.0 ± 0.4 times) as well (Figure 4). For the other uropathogens, the presence of glucose in medium did not significantly stimulate matrix formation.

### 3.3. Metabolic Activity of Uropathogenic Microorganisms

Bacterial metabolic activity was stimulated by NE in all the tested strains, especially at neutral pH 7 (Figure 5). The presence of NE at neutral (7 pH) pH range metabolism was high intensified in *E. coli* (in 3.5 ± 0.2 times)*, Ps. aeruginosa* (in 1.8 ± 0.3 times), *Kl. pneumoniae* (in 2.5 ± 0.3 times), *S. aureus* (in 2.7 ± 0.4 times) and *E. faecalis* strains (in 2.8 ± 0.3 times) in comparison to control (Figure 6). Active stimulation of metabolism in acidic conditions (pH 5) was observed for *Kl. pneumoniae* (in 1.5 ± 0.2 times) and *Ps. aeruginosa* (in 1.8 ± 0.3 times), while in alkaline conditions (pH 8) we observed metabolic activation only in the *E. coli* strains (in 8.0 ± 0.4 times) (Figure 5).

The presence of glucose in PBS also stimulated bacterial metabolism. In acidic pH (5 pH) range metabolism was high intensified in *E. coli* (in 4.0 ± 0.1 times), *Ps. aeruginosa* (in 8.2 ± 0.2 times), *Kl. pneumoniae* (in 4.1 ± 0.2 times) and *E. faecalis* strains (in 1.7 ± 0.4 times) in comparison to control (Figure 6). In neutral pH (7 pH) range in glucose presence metabolism was also activated in *E. coli* (in 1.2 ± 0.2 times), *Ps. aeruginosa* (in 8.4 ± 0.3 times), *Kl. pneumoniae* (in 2.2 ± 0.1 times) and *E. faecalis* strains (in 1.3 ± 0.1 times) in comparison to control (Figure 6). Alkaline conditions, too, allowed glucose to enhance the metabolic activity of *E. coli* (in 2.5 ± 0.1 times) and *E. faecalis* (3.5 ± 0.4 times). For *S. aureus* the metabolic activity showed no significant dependence on the presence of glucose but was actually inhibited in alkaline range (pH 8) in 1.5 ± 0.2 times (Figure 6).

## 4. Discussion

Bacterial infection is the most common type of urinary tract infection (UTI), and can be either episodic or recurrent, uncomplicated or complicated. UTIs affect more than 150 million individuals annually [23]. A severe UTI can lead to urosepsis and septic shock. Some scientists showed a strong correlation between bladder epithelia invasion by uropathogenic bacteria and patients with recurrent UTIs [23]. Intracellular bacteria often recolonise epithelial cells post-antibiotic treatment. Thus, recurrent episodes and active bacterial multiplication in the urinary tract are considered to be important etiological factors for the development of chronic UT diseases. Different studies have described an association between patient stress and susceptibility to infection. The stress hormone NE modulates immunological defense against infection and induces growth in various Gram-negative and Gram-positive bacteria [16,24,25]. In previous studies scientists showed impact of NE on cutaneous Gram-positive skin residents, especially staphylococci [24]. It was shown that NE can accelerate the biofilm formation of *S. epidermidis* and contribute to the competitive behavior of staphylococci. It was shown that *S. epidermidis* suppresses *S. aureus* growth in dual-species biofilms and that NE can modulate this process [24]. In addition, NE can influence virulence factors such as adhesion, greater motility and biofilm production [16]. In this study we considered the effects of pH and soluble substances in urine (NE, glucose) on the proliferation of bacterial cells, matrix formation and their metabolic activity. In our study, we used PBS liquid is a very simple medium solution to imitate urine and to study the effect of the presence of NA and glucose on the activity of uropathogenic bacteria. We refused to use native urine because urine is much more complex and it is a problem to unify urine as a medium for experiments. It is well known, urine can contain various substances, which depends on the quality of the urinary system, lifestyle, nutrition and concomitant non-communicable diseases [1,5,21]. In order to avoid the contribution of all these possible factors affecting the composition of urine, we took PBS as the basic medium and added NE or glucose there. We assessed the contribution of these substances to the metabolic, biofilm and proliferative activity of bacteria. This approach helped us to study the effect of specific components that may be contained in urine and make the most objective conclusions. We also know, that normal urine contains no pathological microorganisms and has a slightly acid pH, but in some cases it may become alkaline, thus it can range in pH from 4.5–8.0 [2]. We therefore chose three value of pH (5, 7 and 8) to determine how pH influenced uropathogenic bacterial strains. We used a PSB with a different pH, which we set before sowing bacteria there, which allowed us to create possible conditions arising in the urinary system if bacteria got there. Naturally, when bacteria multiply, they can change the pH of the medium, which also happens in the human body. However, the pH of the medium can also change under the influence of human metabolic products, which in any case makes this indicator unstable. We set the pH at the initial stage of the experiment and looked at how uropathogenic bacteria would behave if they found themselves in such conditions. The duration of the experiments was 24 h, during which time the bacteria either adapted to these pH values and increased their metabolic activity, or perceived the change in pH as a stress factor and activated the processes of biofilm formation. It is generally assumed that one of the reasons why diabetics are more susceptible to urinary tract infections than non-diabetics is their “sweet urine” [26]. However, very little information is available on this subject. In this study we looked at different combinations of pH with NE and glucose as factors that may promote UTIs. Firstly, we analized the effect of NE on a variety of different strains of uropathogens and showed it promoted increasing biomass production for all species. Similar results had been obtained for *S. aureus* and *E. coli* in our previous study [16]. The presence of glucose also stimulated bacterial growth, especially of the tested strains of *E. coli*, *Ps. aeruginosa*, *E. faecalis* and *S. aureus*. For *Klebsiella* strains significant stimulation by glucose only occurred in acidic conditions. An increase in multidrug-resistant *Klebsiella* spp. strains, responsible for complicated UTIs has been observed by Dobrek et al. [2]. Previous findings showed that high glucose concentrations impair epithelial barrier functions of the UT together with altering cell membrane proteins and cytoskeletal elements, resulting in increasing bacterial burden [21,27].

Matrix production was highly activated by NE and glucose in the *Kl. pneumoniae* strains. This may explain why *Kl. pneumoniae* is one of the most common causative organisms of UTIs [28]. Hyperglycemia is frequent in acute patients due to the increased release of stress hormones such as catecholamines and cortisol, but can also result from the cascade effect of proinflammatory cytokines in emergencies such as acute coronary syndrome, pulmonary edema, pulmonary embolism, injuries, severe infections and sepsis [27,28,29,30]. Thus, all these conditions may result in raised levels of NE and glucose in urine and lead to chronic UTIs caused by *Klebsiella* spp.

The metabolic activity of the Gram-negative uropathogens changed a lot under NE. Thus, *Ps. aeruginosa* and *Kl. pneumoniae* were more active at pH 5 and 7 while *E. coli* was more active at pH 7 and 8. The Gram-positive bacteria showed high metabolic activity under NE only at pH 7. In most cases urine has an acidic pH (close to pH 5–6), so the presence of NE in urine may primarily lead to activation of the Gram-negative pathogens. NE is secreted under conditions of stress in humans. The ability of bacteria to respond to this hormone may have a role in the propagation of infection. Previous investigations have shown significantly elevated urinary NE in diabetic patients with and without UTIs [29,30]. In our study we found that NE and glucose both activated the metabolism of the most frequent uropathogens, such as *E. coli*, *Ps. aeruginosa* and *Kl. pneumoniae*. Our findings are confirmed that, among the drugs associated with an increased risk of development of UTIs, may be immunosuppressants, agents affecting the normal voiding processes together with drugs promoting lithogenesis in the urinary tract or drugs that reduce glucose reabsorption in the kidneys, causing glycosuria [2]. These data can be further used by clinicians in the management of patients with UTI or elevated levels of hormones and glucose in the urine. Based on these data, treatment techniques can be corrected. Due to the spread of antibiotic resistance routine antibiotic therapy can be enhanced by antimicrobial photodynamic therapy (aPDT), which was demonstrated to be effective against various species of uropathogenic bacteria [31]. Thus, if the process is chronicled and there is no effect from antibiotic treatment, an alternative approach based on aPDT with access to the organ cavity through a catheter can be used.

## 5. Conclusions

This study has shown that the pH of urine and the presence of soluble molecules such as NE and glucose may affect biomass production, biofilm formation and the metabolic activity of uropathogenic microorganisms. It was also shown that NE and glucose stimulate biomass production of both Gram-negative (*E. coli*, *Ps. aeruginosa* and *Kl. pneumoniae*) and Gram-positive uropathogens (*S. aureus* and *E. faecalis*), the metabolic activity (*E. coli*, *Ps. aeruginosa* and *Kl. pneumoniae*) and matrix production in *Kl. pneumoniae* strains. Thus, NE and glucose in urine may stimulate uropathogenic bacteria to multiply in the urinary tract and lead to persistent infection, especially when the patient is under stress or diabetic.

## Figures and Tables

**Figure 1 microorganisms-11-00862-f001:**
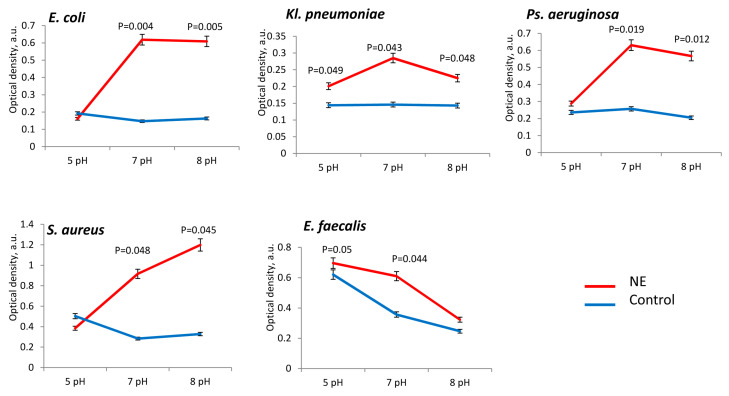
Biomass production of uropathogenic microorganisms is influenced by pH and the presence of NE in PBS.

**Figure 2 microorganisms-11-00862-f002:**
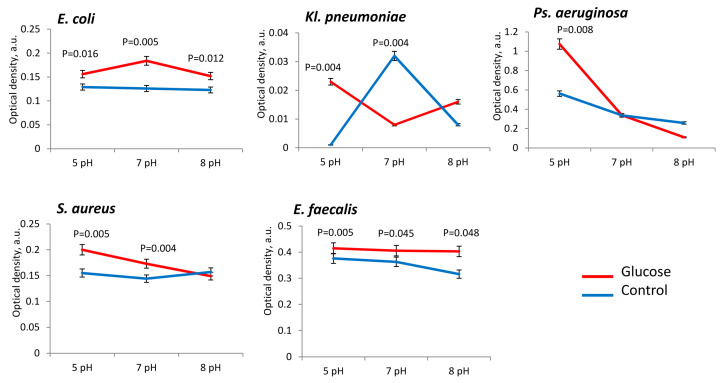
Biomass production of uropathogenic microorganisms is influenced by pH and the presence of glucose in PBS.

**Figure 3 microorganisms-11-00862-f003:**
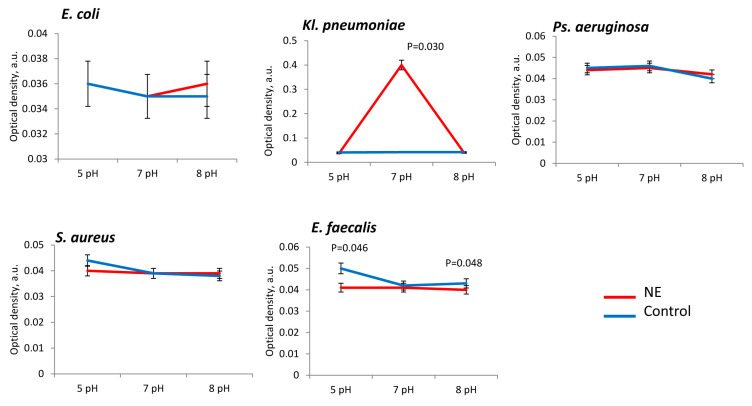
Matrix production of uropathogenic microorganisms is influenced by pH and the presence of NE in PBS.

**Figure 4 microorganisms-11-00862-f004:**
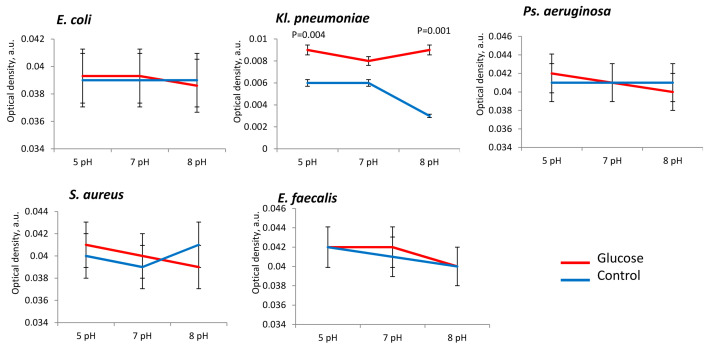
Matrix production of uropathogenic microorganisms is influenced by pH and the presence of glucose in PBS.

**Figure 5 microorganisms-11-00862-f005:**
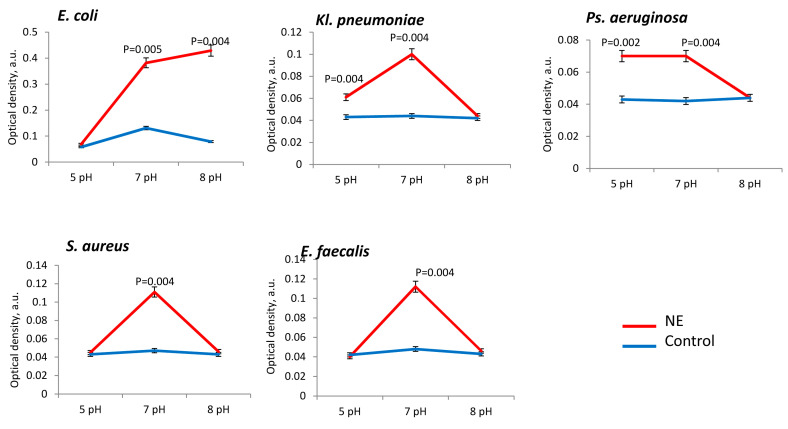
Bacterial metabolic activity is influenced by pH and the presence of NE in PBS.

**Figure 6 microorganisms-11-00862-f006:**
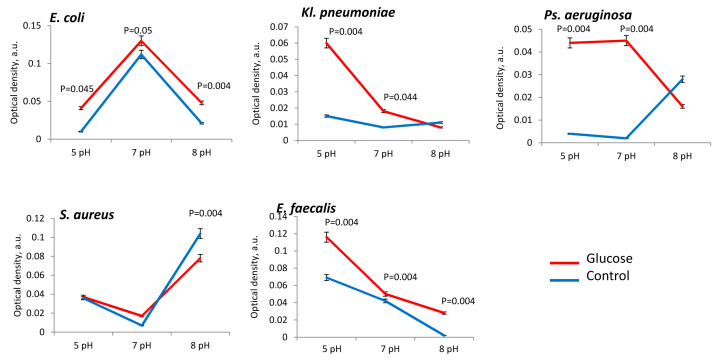
Bacterial metabolic activity is influenced by pH and the presence of glucose in PBS.

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
