# Peer review of "Effect of pH, Norepinephrine and Glucose on Metabolic and Biofilm Activity of Uropathogenic Microorganisms"

_microorganisms, 2023, doi:10.3390/microorganisms11040862_

Round 1
Reviewer 1 Report
Dear authors,
After the review process, I have several comments: in the abstract should be included more numerical data; all sections from Materials and Methods should have references; the paper has a discussion section without clear comments about the future valorization of the results; you should make a connection with past alternative studies (in vitro, for example); also, you should include comments that present novel insights to prevent and treat recurrent urinary tract infections.
Best regards!
Author Response
First of all, we would like to thank editorial board member and reviewers for the very useful comments which allow improving the quality of the manuscript. All changes in the manuscript are highlighted by yellow.
Reviewer 2 Report
The authors of “Effect of pH and soluble components of urine on metabolic and 2 biofilm activity of uropathogenic microorganisms” set out to determine if the pH, norepinephrine and glucose have an effect on the various uropathogens and their ability to produce biofilms. Overall, the paper is well written. However, I do have concerns about the conclusions being made since the experiments were very simple and performed in PBS and not urine itself. Urine can be a very complex medium so I appreciate the ability of these authors to begin to deconstruct the urine to simple soluble components to test their influence individually on each uropathogen. However, throughout the text they change their language from “in the medium” to “in the urine” making it appear that these changes are occurring in urine when they are not. In addition, was the pH maintained throughout the experiment or is the pH an indication of the media prior to the monitoring the growth of the bacteria? This is a question that I need answered before I could consider this article for publication. Below are other edits and questions that I have for the document:
Line 12-Space needed between the words “are” and “mainly”
Line 18-Should have the genus name fully written since it is the first appearance in the abstract
Line 22-Gram-positive and Gram-negative need a capital “G”
Line 89-90- PBS liquid is a very simple medium solution and does not imitate urine and urine is much more complex. Unless there is a publication out there that states this, would you be able to provide that?
Line 97-100-Was the pH maintained throughout the experiment or is the pH an indication of the media prior to the monitoring the growth of the bacteria? If so, how was that done and can you please explain that in the methods section here? I can imagine that as the pH would change as the bacteria are growing in the presence of or glucose and NE. If it was just the starting pH, can you please clarify that in the methods?
Line 144- Italicize the bacterial names
Line 145-6- Italicize the bacterial names
Figure 1 and 2- Do you have separate growth curves of these organisms in this medium? That may be a more helpful depiction on how these organisms are growing in the presence of glucose and norepinephrine instead of just the endpoint measurements. Again, was the pH maintained throughout the experiments or was it just at the beginning? Because as bacteria grow in media they can change the pH of the medium.
Line 154- Italicize the bacterial names
Line 154-156- This conclusion is a bit of a reach since you are not actually monitoring the growth of these organisms in urine but instead in a simple PBS solution.
Line 156-Is this actually alkaline urine or alkaline medium? Please clarify which one. If it is urine, that was not described in the methods and needs to be added.
Line 157- Italicize the bacterial names
Line 164- Italicize the bacterial name
Line 166- Italicize the bacterial names and Gram-positive needs a capital “G”
Line 173- Italicize the bacterial name
Line 189-This is not in urine, it is a PBS medium, please correct to be consistent with your previous paragraphs stating “presence of glucose in the medium”
In all Figures, can we get an actual p-value for all of the values that were statistically significant displayed within the figures? Also you need to change the language in all of the figures from “in urine” to “in medium” since you are not using urine.
Line 205-206-Gram-positive and Gram-negative need a capital “G”
Line 235- Gram-negative needs a capital “G”
Line 237- Gram-positive needs a capital “G”
Line 239- Gram-negative needs a capital “G”
Line 253- Gram-negative needs a capital “G”
Line 254- Gram-positive needs a capital “G”
Author Response

(The authors gave the same response as above.)

Round 2
Reviewer 1 Report
Dear authors,
You do not respond directly to my comments, especially about in vitro studies. You should rewrite the valorization of the paper and what are the negative points of the paper.
Best regards!
Author Response
Thank you for your comments you helped us to do best. Our replies are in the attachment. All changes in the manuscript are highlighted by yellow.

Reviewer 2 Report
I appreciate the authors taking the time to address the edits/suggestions/concerns that I had in the initial submission. The document is much stronger now and I approve of it being published in its current form.
Author Response
Thank you for your decision you helped us to do best.